# PKCα Inhibition as a Strategy to Sensitize Neuroblastoma Stem Cells to Etoposide by Stimulating Ferroptosis

**DOI:** 10.3390/antiox10050691

**Published:** 2021-04-28

**Authors:** Lorenzo Monteleone, Andrea Speciale, Giulia Elda Valenti, Nicola Traverso, Silvia Ravera, Ombretta Garbarino, Riccardo Leardi, Emanuele Farinini, Antonella Roveri, Fulvio Ursini, Claudia Cantoni, Maria Adelaide Pronzato, Umberto Maria Marinari, Barbara Marengo, Cinzia Domenicotti

**Affiliations:** 1General Pathology Section, Department of Experimental Medicine, University of Genoa, 16126 Genoa, Italy; lorenzo.monteleone@edu.unige.it (L.M.); giuliaelda.valenti@edu.unige.it (G.E.V.); nicola.traverso@unige.it (N.T.); claudia.cantoni@unige.it (C.C.); maidep@unige.it (M.A.P.); umm@unige.it (U.M.M.); 2Mutagenesis and Cancer Prevention Unit, IRCCS Ospedale Policlinico San Martino, 16132 Genoa, Italy; andrea.speciale@hsanmartino.it; 3Human Anatomy Section, Department of Experimental Medicine, University of Genoa, 16126 Genoa, Italy; silvia.ravera@unige.it; 4Department of Cardiovascular Medicine, IRCCS Humanitas Clinical and Research Center, Rozzano, 20089 Milan, Italy; ombretta.garbarino@humanitasresearch.it; 5Department of Pharmacy, University of Genoa, 16126 Genoa, Italy; riclea@difar.unige.it (R.L.); farinini@difar.unige.it (E.F.); 6Department of Molecular Medicine, University of Padua, 35122 Padua, Italy; antonella.roveri@unipd.it (A.R.); fulvio.ursini@unipd.it (F.U.); 7Laboratory of Clinical and Experimental Immunology, Integrated Department of Services and Laboratories, IRCCS Istituto G. Gaslini, 35122 Genoa, Italy

**Keywords:** cancer stem cells, chemoresistance, glutathione, lipid peroxidation, ZEB-1, GPX4, ferroptosis

## Abstract

Cancer stem cells (CSCs) are a limited cell population inside a tumor bulk characterized by high levels of glutathione (GSH), the most important antioxidant thiol of which cysteine is the limiting amino acid for GSH biosynthesis. In fact, CSCs over-express xCT, a cystine transporter stabilized on cell membrane through interaction with CD44, a stemness marker whose expression is modulated by protein kinase Cα (PKCα). Since many chemotherapeutic drugs, such as Etoposide, exert their cytotoxic action by increasing reactive oxygen species (ROS) production, the presence of high antioxidant defenses confers to CSCs a crucial role in chemoresistance. In this study, Etoposide-sensitive and -resistant neuroblastoma CSCs were chronically treated with Etoposide, given alone or in combination with Sulfasalazine (SSZ) or with an inhibitor of PKCα (C2-4), which target xCT directly or indirectly, respectively. Both combined approaches are able to sensitize CSCs to Etoposide by decreasing intracellular GSH levels, inducing a metabolic switch from OXPHOS to aerobic glycolysis, down-regulating glutathione-peroxidase-4 activity and stimulating lipid peroxidation, thus leading to ferroptosis. Our results suggest, for the first time, that PKCα inhibition inducing ferroptosis might be a useful strategy with which to fight CSC chemoresistance.

## 1. Introduction

Neuroblastoma (NB), a pediatric cancer originating from the neural crest cells, accounts for about 6% of all cancers in children [1]. The development of the neural crest is correlated with the epithelial–mesenchymal transition (EMT) [2,3], a phenomenon that is strictly related to MYCN amplification [4], a negative prognostic factor that characterizes advanced tumor stages and aggressive progression of NB [5,6].

Several studies have reported that the genes involved in the EMT process are also implicated in regulating cancer stem cell (CSC) generation and propagation [7]. CSCs are a limited cell population found in different tumors, including NB [8], and are involved in various stages of the tumorigenic process and in cancer response to therapy [9,10,11]. In this regard, it has been demonstrated that CSCs, having physiologically low levels of reactive oxygen species (ROS) and high antioxidant defenses in comparison to cancer cells, are strongly involved in chemoresistance [12]. The maintenance of these low ROS levels is due to the ability of CSCs to synthesize high amounts of glutathione (GSH) by over-expressing xCT, a membrane transporter able to modulate the intracellular levels of cysteine, the limiting amino acid involved in GSH synthesis [13]. Moreover, it has also been found that CSCs express high levels of CD44, a well-known stemness marker [14] that is able to induce xCT membrane stabilization [15] and whose expression is modulated by protein kinase C α (PKCα) [16,17].

Based on these considerations, anti-tumor research should be focused on discovering innovative approaches able to limit GSH availability in cancer cells and, in particular, in CSCs. In fact, it has been shown that L-Buthionine-sulfoximine (BSO), a GSH-depleting agent, sensitizes NB cells to chemotherapeutic drugs [18,19,20,21] and CSCs to radiotherapy [13], by inducing ROS over-production. However, the clinical use of BSO is not actually feasible due to the secondary toxic effects of this compound, in addition to those resulting from the use of traditional radiation and drug therapies [22].

Therefore, in order to investigate new nontoxic therapeutic approaches, both Etoposide-sensitive and Etoposide-resistant NB-derived CSCs were long-term treated with Etoposide, alone or in combination with sulfasalazine (SSZ), or with an inhibitor of the PKCα-dependent pathway (C2-4, a peptide inhibitor), which can target xCT directly or indirectly, respectively.

The herein results demonstrate that both combinations of Etoposide with SSZ or C2-4 efficiently counteract the propagation of Etoposide-resistant CSCs by preventing EMT transition and by triggering ferroptosis via down-regulation of GPX4 activity.

## 2. Materials and Methods

### 2.1. Chemicals

Etoposide was purchased from Calbiochem (Merk KGaA, Darmstadt, Germany) and xCT inhibitor sulfasalazine (SSZ) and PKCα peptide inhibitor (C2-4) from Cayman Chemical (Ann Arbor, MI, USA). The stock solutions of these compounds were prepared using dimethyl sulfoxide (DMSO, Sigma-Aldrich, St. Louis, MO, USA) as a solvent.

### 2.2. Cancer Stem Cell (Csc) Generation

To select neurospheres (3D cultures), floating parental cells derived from 2D cultures of HTLA-230 and HTLA-ER [23] cells were harvested, centrifuged (117 rcf × 8 min), seeded (16 × 10^4^) and grown in DMEM-F12 Knock-out (Life Technologies, Carlsbad, CA, USA) containing 1% penicillin/streptomycin (Euroclone S.p.A., Pavia, Italy), 2% B27 (Life Technologies), 40 ng/mL basal growth factor for fibroblasts (bFGF) (R&D Systems, Inc., Minneapolis, MN, USA) and 20 ng/mL epidermal growth factor (EGF) (Life Technologies) [24]. It is important to outline that the expression of stem cell markers was evaluated (see Section 2.4) in order to verify the staminality and, only after this characterization, were the neurospheres renamed as CSCs. The HTLA-CSCs and ER-CSCs were split once a week to favor their propagation and, at each passage, the culture medium consisted of 50% fresh medium and 50% of the medium in which the cells were grown [24]. In order to analyze CSC propagation, at any split, the CSCs were collected, centrifuged, mechanically dissociated in PBS and then counted using a Burker chamber (Marienfeld, Germany).

Parental HTLA-230 and HTLA-ER cells were maintained in RPMI 1640 medium (Euroclone) containing 10% fetal bovine serum (FBS; Euroclone), 2 mM glutamine (Euroclone), 1% penicillin/streptomycin (Euroclone), 1% sodium pyruvate and 2% amino acid solution (Sigma-Aldrich, St. Louis, MO, USA). All cell lines were periodically tested with semi-quantitative PCR for the evaluation of any mycoplasma contamination (Mycoplasma Reagent Set; Euroclone).

### 2.3. Treatments

Both CSC populations were treated once a week with 1 mM BSO or three times a week with 0.1 µM C2-4, with 5 µM SSZ or with 1.25 µM Etoposide, administered once a week alone or in combination with C2-4 or SSZ. The treatments were protracted for 6 weeks and stock solutions (1 mM C2-4, 50 mM SSZ and 50 mM Etoposide) were prepared by dissolving each drug in DMSO. In order to demonstrate that the doses of DMSO used to dissolve the compounds did not affect the subsequent analyses, pilot studies were performed in parallel with the experiments.

### 2.4. Immunofluorescence and Cytofluorimetric Analysis

In order to investigate the expression of CD44, Oct3/4, CD44v9, xCT, and vimentin (used as a control of cell permeabilization), cells were analyzed by flow cytometry. In detail, both untreated CSC populations were collected, centrifuged (117 rcf × 8 min), and then re-suspended in the TrypLE ™ Express Enzyme 1X solution (Invitrogen, Carlsbad, CA, USA). After monitoring CSC disaggregation by microscope analysis, cells were diluted in PBS (137 mM NaCl, 2.6 mM KCl, 10 mM Na_2_HPO_4_, 1.8 mM KH_2_PO_4_, pH 7.4). For intracellular staining, prior to incubation with primary mAbs, cells were permeabilized and fixed for 30 min at 4 °C with the BD Cytofix/Cytoperm ™ kit (BD Biosciences, Franklin Lakes, NJ, USA). After this step, CSCs were re-suspended in 1% PBS/Saponin solution (Sigma-Aldrich, Milano, Italy) and then centrifuged (365 rcf × 10 min). Subsequently, CSCs were incubated for 30 min at 4 °C with primary antibodies: rabbit polyclonal anti-CD44 (Cosmo Bio, Tokio, Japan), rat monoclonal anti-Oct3/4 (EM92, PE-linked; eBioscience, San Diego, CA, USA), rat monoclonal anti-CD44v9 (RV3, Cosmo Bio), rabbit polyclonal anti-xCT (H-121, Santa Cruz Biotechnology, Dallas, TX, USA), and mouse monoclonal anti-vimentin (V9, Santa Cruz Biotechnology).

Labelled cells were then washed with 2% FBS solution in PBS and incubated for 30 min at 4 °C with the appropriate isotype-matched secondary antibodies: goat anti-rabbit-IgG-PE (Southern Biotech, Birmingham, AL, USA), goat anti-rat-IgG-FITC (Sigma-Aldrich), goat anti-mouse-IgG-PE (Southern Biotech). Finally, all samples were fixed with PBS/1% formaldehyde and analyzed with the FACScalibur flow cytometer (BD Biosciences). Data were analyzed by the CellQuest software (BD Biosciences).

### 2.5. HPLC Analysis of Intracellular Glutathione Levels

Intracellular levels of reduced glutathione (GSH) and oxidized glutathione (GSSG) were assessed by high performance liquid chromatography (HPLC) using the methods reported by Reed for total GSH [25] and Asensi for GSSG quantification [26].

Untreated and treated CSCs (1 × 10^6^ cells) were collected and centrifuged (117 rcf × 8 min). Then, the pellets were washed in PBS, precipitated with 10% perchloric acid (PCA) and the thiol groups were blocked with iodacetic acid (alkaline pH 8–9, 10 min, room temperature, in the dark). Subsequently, the analytes were converted to 2,4-dinitrophenyl derivatives by incubating the samples with 1% 1-Fluoro-2,4-dinitrobenzene (FDNB) at 4 °C in the dark overnight. The quantitative determination of the derivatized analytes was carried out by HPLC equipped with an NH2 Spherisorb column and a UV detector set at 365 nm with a flow rate of 1.25 mL/min. The mobile phase was maintained at 80% solution A (80% methanol in water) and 20% solution B (0.5 M sodium acetate in 64% methanol in water) for 5 min, followed by a 10-min-linear gradient to 1% A and 99% B. The chromatography was performed with gradient elution.

In order to evaluate GSSG, the samples were incubated with 20 mM N-ethyl-maleimide (NEM) in PCA and, after precipitation and alkalization of the sample (alkaline pH 8–9, 10 min, room temperature, in the dark), derivatization was carried out by adding 1% FDNB. In order to allow the elution of GSSG, the mobile phase was kept at 99% solution B for a further 15 min.

The data obtained was normalized by the intracellular amount of protein and expressed as µEq/mg of protein.

### 2.6. ATP Synthesis

The ATP content was measured using the luciferin/luciferase ATP bioluminescence assay kit CLSII (Roche, Basel, Switzerland) on a Luminometer (Triathler, Bioscan, Washington, DC, USA) [23]. ATP standard solutions (Roche, Basel, Switzerland) in the concentration range of 10^−10^–10^−7^ M were used for calibration. Assay was carried out at 37 °C over 2 min by measuring formed ATP from added ADP. Untreated and treated CSCs were incubated for 10 min with the assay buffer (10 mM Tris-HCl pH 7.4, 50 mM KCl, 1 mM EGTA, 2 mM EDTA, 5 mM KH_2_PO_4_, 2 mM MgCl_2_, 0.6 mM Ouabain, 0.040 mg/mL Ampicillin, 0.2 mM di(adenosine-5′) penta-phosphate, 0.2 mM and 5 mM pyruvate plus 2.5 mM malate). Afterwards, ATP synthesis was induced by the addition of 0.3 mM ADP.

### 2.7. Oxygen Consumption Rate (OCR)

In order to measure the respiratory activity, 2 × 10^5^ cells were analyzed by using an amperometric electrode for O2 placed in an isolated chamber and stirring maintained at 37 °C, as previously reported [23]. Untreated and treated CSCs were collected and centrifuged (117 rcf × 8 min). Then, the pellets were suspended in the assay buffer (137 mM NaCl, 5 mM KH_2_PO_4_, 5 mM KCl, 0.5 mM EDTA, 3 mM MgCl_2_ and 25 mM Tris–HCl, pH 7.4), and permeabilized with 0.3% digitonin for 10 min. Then, the sample was transferred to the chamber; so as to measure the maximum respiration rate, 5 mM pyruvate plus 2.5 mM malate were added.

### 2.8. Glucose Consumption

Glucose consumption was assessed by measuring its concentration using a double beam spectrophotometer (UNICAM UV2, Analytical S.n.c., Langhirano, PR, Italy), by the hexokinase (HK) and glucose 6 phosphate dehydrogenase (G6PD) coupling system, following the reduction of NADP at 340 nm. [23]. The buffer assay contained 100 mM Tris HCl, pH 7.4, 2 mM ATP, 10 mM NADP, 2 mM MgCl_2_, 2 IU of HK and 2 IU of G6PD. The reaction was started after the addition of 5 μL of CSC medium.

### 2.9. Lactate Release

The concentration of lactic acid released by CSCs in the culture medium was analyzed by spectrophotometric analysis as previously reported [23]. The assay buffer contained 100 mM Tris/HCl (pH 8), 5 mM NAD+ and 1 IU/mL of lactate dehydrogenase. Samples were analyzed before and after the addition of 4 μg of purified lactate dehydrogenase.

### 2.10. MDA Production

Malondialdehyde (MDA) levels were analyzed by thiobarbituric acid assay (TBARS) as previously reported [23,27]. Briefly, 50 µg of total proteins, derived from each sample, were dissolved in 300 µL of Milli-Q water and added to 600 µL of TBARS solution containing 15% trichloroacetic acid (TCA) and 26 mM thiobarbituric acid (TBA) in 0.25 N HCl. This mixture was incubated for 40 min at 100 °C and then centrifuged (20,800 rcf × 2 min). Subsequently, the supernatant was collected and analyzed on the spectrophotometer using a wavelength of 532 nm. In order to calculate MDA concentration, the absorbance of each sample was compared with that obtained from a standard curve evaluated with known concentrations of MDA (0.75, 1, and 2 µM).

### 2.11. ROS Production

Detection of ROS levels was evaluated by incubating CSCs with 5µM 2′-7′ dichlorofluorescein-diacetate (DCFH-DA; Sigma-Aldrich) as previously reported [23].

### 2.12. Western Blot Analyses

Immunoblots were carried out according to standard methods [28] using rabbit antibody anti-N-Cadherin (D4R1H), anti-ZEB-1 (D80D3), anti-Vimentin (D21H3), anti-β-Catenin (D10A8), anti-Claudin-1 (D5H1D) (Cell Signaling Technology Inc., Danvers, MA, USA Upstate, Lake Placid, NY, USA) and anti-GPX4 (Abcam, Cambridge, UK). Anti-rabbit secondary antibodies coupled with horseradish peroxidase (Cell signaling Technologies) were utilized.

### 2.13. GPX4 Activity

GPX4 activity was evaluated by standard methods [29]. In detail, cell pellets were re-suspended in 0.75 mL lysis buffer (0.1 M Tris-HCl, 0.25 M sucrose, protease inhibitors, pH 7.5) and then sonicated and used in the test (0.1–0.2 mL of sample per test). Samples were mixed with the assay buffer (0.1 M Tris-HCl pH 7.8, 5 mM EDTA, 5 mM GSH, 0.1% (*v/v*) Triton X-100, 0.16 mM NADPH and 0.6 IU/mL Glutathione Reductase (GR)) and incubated for 5 min at 25 °C. Then the baseline was recorded at 340 nm for about 1 min and finally the enzymatic activity was started by adding phosphatidylcholine hydroperoxide (0.020 mM). The quantification of the activity was done on the basis of the net speed with which the absorbance decreases after the addition of the substrate (net speed = speed after the addition of the substrate − baseline speed).

### 2.14. Principal Component Analyses (PCA)

PCA is a data display method for multivariate data. Given a data set in which each sample is described by n variables, PCA aims to find new directions, linear combinations of the original ones [30].

The first component (PC1) corresponds to the direction explaining the maximum variance, while PC2 is the direction, orthogonal to PC1, explaining the maximum variance not explained by PC1, and so on. The result of such a transformation is that a limited number of components is sufficient to explain the relevant part of the information.

The loadings are the coefficients of the linear combinations corresponding to the PCs. By plotting them in a loading plot it is possible to understand the relationships among the variables in the multivariate space.

On the other side, the score plot (the scores being the coordinates of the samples in the new space defined by the PCs) allows visualization of the location of the samples in the space described by the PCs, hence making possible to check similarities and differences among the samples. The elaborations and the plots were carried out by using CAT software [31].

### 2.15. Statistical Analysis

Results were expressed as mean ± SEM from at least four independent experiments. The statistical significance of parametric differences among the sets of experimental data was evaluated by one-way ANOVA and Dunnett’s test for multiple comparisons.

## 3. Results and Discussion

### 3.1. Neurospheres, Isolated from Parental HTLA-230 and HTLA-ER Cells, Are Cancer Stem Cells (CSCs) and GSH Plays a Crucial Role in Their Generation and Propagation

In order to verify if the floating neurospheres, isolated from HTLA-230 and HTLA-ER cells [23], were formed by stem cells, the expression of CD44 and Oct-3/4, known markers of stemness, was analyzed. As shown in Figure 1a, isolated neurospheres expressed both CD44 and Oct-3/4, demonstrating that both cell populations have staminality characteristics and, consequently, have been referred to as HTLA- and ER-cancer stem cells (HTLA-CSCs and ER-CSCs).

Since the presence of CSCs [12] and glutathione (GSH) play a crucial role in chemoresistance [23], both CSC populations were chronically treated (once a week for 6 weeks) with 1 mM BSO in order to investigate whether a relationship between GSH levels and stemness exists.

As shown in Figure 1b, BSO markedly counteracted the formation of HTLA-CSCs after two weeks of treatment, while a similar effect was observed in ER-CSCs after five weeks. These results demonstrate that GSH plays a crucial role in the formation and maintenance of CSCs and suggest that its depletion could be employed to increase sensitivity of CSCs to therapeutic approaches.

However, although BSO has been found to counteract CSC formation, both in vitro and in vivo [32,33,34], and has been included in NB clinical trials [35], its therapeutic use is limited by its short half-life and by its non-selective depleting effect on neoplastic cells [22]. In order to overcome this limitation, an effective strategy could be the indirect modulation of GSH levels by acting on GSH-related targets. In this context, it has been reported that CD44, a known stemness membrane marker [17,36], and, in particular, its variant 9 (CD44v9) contributes to the stabilization of xCT, an anti-port protein involved in the uptake of cystine, an amino acid that is essential for GSH biosynthesis. Considering that the expression of CD44 is modulated by protein kinase C (PKC)α [17,36], an opportunity would be to reduce the CD44 expression by inhibiting this kinase. As shown in Figure 1c, both CSC populations expressed CD44v9 and xCT, thus justifying the use of C2-4, a PKCα inhibitor, or of sulphasalazine (SSZ), a xCT inhibitor, in the reported studies.

### 3.2. Etoposide Prevents the Formation of HTLA-CSCs after 3 Weeks of Treatment While It Completely Counteracts the Formation of ER-CSCs when It Is Combined with SSZ or C2-4 for 6 Weeks

By analyzing HTLA-CSCs, it was observed that Etoposide reduced the propagation of the formed CSCs by 35% after only one week of exposure and totally prevented CSC formation after three weeks (Figure 2a). Moreover, similar effects were obtained in CSCs co-treated with either Etoposide and SSZ (Figure 2a, left panel) or Etoposide and C2-4 (Figure 2a, right panel). However, treatments with SSZ (Figure 2a, left panel) or C2-4 alone (Figure 2a, right panel) did not significantly alter this parameter during the analyzed time points.

In contrast, the same treatments did not induce any significant alteration in the formation/propagation of ER-CSCs until two weeks of exposure (Figure 2b). In fact, Etoposide treatment reduced CSC propagation by 38% at four weeks, and this effect remained unchanged until six weeks of treatment (Figure 2b). By analyzing the results obtained in co-treated CSCs, it was observed that Etoposide combined with SSZ or Etoposide combined with C 2-4 reduced the ER-CSC propagation by 30% and 55%, respectively, after three weeks and totally prevent the formation of CSCs after six weeks (Figure 2b). Moreover, as shown in Figure 2b left panel, the results obtained in ER-CSCs, treated with SSZ alone for six weeks, were comparable to those found in CSCs exposed to only Etoposide, whereas the treatment with C2-4 did not affect CSC formation and propagation (Figure 2b, right panel).

Since the formation of HTLA-CSCs was totally inhibited after three weeks of treatments and that of ER-CSCs after six weeks, the following analyses were performed after two and five weeks respectively.

Together these results demonstrate that ER-CSCs maintain their resistance to Etoposide and show that ER-CSC generation is totally inhibited by co-treatments with SSZ or C2-4. Moreover, the data obtained confirm that the inhibition of xCT is a useful strategy for enhancing the effect of chemotherapy [37,38,39,40,41,42] and, although the use of SSZ is limited due to its low bioavailability, its employment has been recently considered to treat high risk NB [43]. In addition, the results obtained in C2-4-co-treated cells suggest that the inhibition of PKCα could be a novel way to sensitize CSCs to therapy by sustaining GSH depletion. In this regard, several clinical trials (NCT00042679; NCT00003236; NCT00017407) have been carried out using Aprinocarsen, an anti-sense oligonucleotide that targets PKCα mRNA, even if, unfortunately, the results obtained are not encouraging [44]. However, a large number of studies have reported the efficacy of PKCα inhibitors in sensitizing cancer cells to chemotherapeutic drugs [45,46,47,48].

### 3.3. Etoposide Induces a Marked Depletion of GSH in HTLA-CSCs after Two Weeks While the Same Effect Is Not Observed in ER-CSCs until Five Weeks of Treatment

Untreated HTLA-CSCs and ER-CSCs displayed comparable GSH levels (about 17.07 μEq/g and 17.37 µEq/g proteins, respectively). After two weeks of Etoposide exposure, an 80% reduction in GSH was observed in HTLA-CSCs with respect to untreated CSCs (Figure 3a). At the same time, both co-treatment conditions (C2-4 + Etoposide or SSZ + Etoposide), exerted similar effects to those detected in Etoposide-treated CSCs (Figure 3a), while C2-4 or SSZ alone did not significantly alter the GSH amount (Figure 3a). Notably, the same two-week treatments in ER-CSCs did not induce any significant changes in GSH levels (Figure 3b).

After five weeks, untreated HTLA-CSCs and ER-CSCs displayed a different GSH content of 4.7 μEq/g and 8.99 μE/g protein, respectively. Since the five-week-treatment with Etoposide, alone or in combination, totally counteracted HTLA-CSC formation (Figure 2a), it was not possible to analyze GSH. As shown in Figure 3c, Etoposide decreased GSH of ER-CSCs by 60% in comparison to untreated CSCs, and co-treatments, with C2-4 or SSZ, which did not modify the GSH-depleting effect of Etoposide.

The amount of GSSG, the oxidized form of GSH, was below the detection limits in both CSC populations after two weeks while it was depleted by 60–65% in ER-CSCs after the five-week-treatment with Etoposide, alone or in combination with C2-4 or SSZ (data not shown).

These results demonstrate that five-week treatment with Etoposide markedly reduces GSH levels in ER-CSCs and that this effect is only due to the action of Etoposide, given that a similar loss of GSH levels is observed in co-treated CSCs and neither C2-4 nor SSZ, alone or combined with Etoposide, modulate GSH levels.

This data demonstrates that the survival of Etoposide-resistant CSCs is not totally dependent on GSH since, even though Etoposide exerts a severe GSH-depleting action, it does not totally counteract CSC generation, suggesting that other factors might contribute to the maintenance of cancer stem cell survival. In this regard, it has been widely demonstrated that cancer cells usually depend on glycolysis for energy production [49,50] while chemoresistant cancer cells [23,51,52] and CSCs undergo a metabolic rewiring, stimulating OXPHOS with a major ATP production [53,54,55,56]. Therefore, based on these findings, the metabolic profile of untreated and treated CSCs was analyzed.

### 3.4. ER-CSCs Maintain an Efficient OXPHOS Metabolism Which Is Impaired Only by Co-Treatments

As shown in Figure 4a, ATP synthesis, detected in untreated HTLA-CSCs, after two weeks, was doubled in respect of that of ER-CSCs. In HTLA-CSCs, two weeks of Etoposide, alone or in combination with C2-4 or SSZ, reduced ATP production by 70% in respect of untreated CSCs (Figure 4a). Notably, in ER-CSCs, two weeks of Etoposide was ineffective and only co-treatment of Etoposide with SSZ reduced ATP synthesis by 25% compared to both untreated and Etoposide treated CSCs (Figure 4b).

After five weeks, untreated CSC populations had comparable levels of ATP (about 10 nmol/min/1 × 10^6^ cells). Since five weeks of treatment with Etoposide, alone or in combination, totally counteracted HTLA-CSC formation (Figure 2a), the evaluation of ATP synthesis was not possible while in ER-CSCs, ATP decreased by 10% after Etoposide and by 35% by the co-treatments (Figure 4c).

Moreover, in HTLA-CSCs, two weeks of Etoposide treatment reduced the oxygen consumption rate (OCR) by 75% in respect of untreated CSCs (Figure 5a) and similar results were obtained in co-treated ones (Figure 5a). At the same time point, in ER-CSCs, the OCR was not modified either by Etoposide or co-treatments and it was increased by 15% and 35% only by C2-4 or SSZ respectively (Figure 5b). Since the five-week-treatment with Etoposide, alone or in combination, totally counteracted HTLA-CSC formation (Figure 2a), OCR analysis was not possible. Notably, in ER-CSCs, none of the treatments significantly altered the OCR (Figure 5c).

To verify the efficiency of OXPHOS, the P/O value was calculated. In detail, in both CSC populations, the P/O ratio was 2.3 ± 0.2, which is comparable to the physiological level of 2.5 as reported by Hinkle [57] and interestingly, only SSZ, administered alone or in combination with Etoposide for two or five weeks, led to a P/O value of 1.8 ± 0.15.

In order to complete the analysis of cell metabolism, glucose consumption and lactate release were evaluated. As shown in Figure 6a, untreated HTLA-CSCs consumed about 40% more glucose than untreated ER-CSCs. In HTLA-CSCs, two-week-Etoposide exposure increased glucose consumption by 60% in respect of untreated CSCs and the co-treatments did not modify the Etoposide-induced effect (Figure 6a). In ER-CSCs, treatments with either Etoposide, C2-4 or SSZ increased glucose consumption by 30% as compared to untreated CSCs, and the co-treatments with C2-4 or SSZ contributed to stimulating this parameter by a further 60% and 25%, respectively (Figure 6b).

Since five-week-treatment with Etoposide, alone or in combination, totally counteracted HTLA-CSC formation (Figure 2a), glucose consumption analysis was not possible. Noteworthy, in ER-CSCs, the trend observed after five weeks was similar to that observed after two weeks and, also in this case, the co-treatments stimulated the Etoposide-induced effect on glucose consumption (Figure 6c).

As expected, in line with the highest level of glucose consumption observed in untreated HTLA-CSCs, an increase of 50% in the lactate release was detected in comparison with untreated ER-CSCs (Figure 7a,b). After two weeks, the Etoposide exposure of HTLA-CSCs stimulated the lactate release by 80% in respect of untreated CSCs, and the co-treatments did not modify the Etoposide-induced effect (Figure 7a). In ER-CSCs, a similar result was obtained only in the presence of C2-4 or SSZ, given alone or in combination with Etoposide, while Etoposide per se did not induce any effect (Figure 7b). Interestingly, in ER-CSCs, after five weeks, the co-treatments with C2-4 or SSZ stimulated lactate release by 30% in respect of both untreated and Etoposide-treated ER-CSCs (Figure 7c).

Taken together, the metabolic findings indicate that, although both untreated CSC populations have an efficient OXPHOS metabolism, HTLA-CSCs, following Etoposide exposure, activate glycolysis while ER-CSCs maintain their metabolic adaptation and are able to choose glycolytic metabolism after five-week-co-treatments with C2-4 or SSZ. This data, besides confirming the propensity of CSCs to obtain energy by stimulating OXPHOS [58], also demonstrates that this metabolic rewiring is reversible [59] and that, in having a crucial role in CSC survival, it could be targeted in order to eradicate chemoresistant CSC populations. In agreement with these findings, OXPHOS inhibitors (e.g., metformin) have been found to increase therapy sensitivity of cancer stem cells [60,61].

### 3.5. Co-Treatments Are Able to Induce Lipid Peroxidation in ER-CSCs

In order to better characterize cell damage, the production of malondialdehyde (MDA), a marker of membrane lipid peroxidation, was evaluated in both CSC populations. As observed in Figure 8a, two-week-treatments of HTLA-CSCs with Etoposide or SSZ alone increased MDA levels by 100% and the co-treatment with SSZ increased this parameter by a further 30% (Figure 8a). Notably, in ER-CSCs, only two-week-exposure to SSZ, alone or in combination with Etoposide, induced an 85% increase in MDA production (Figure 8b). After five weeks, in ER-CSCs, all treatments maintained MDA at similar levels to those observed after two weeks. In addition, a 35% increase in MDA levels was found in C2-4-co-treated ER-CSCs (Figure 8c).

Considering that mitochondrial oxidative phosphorylation can generate ROS, this parameter was evaluated in both CSC populations.

As shown in Figure 9a, after two-week-exposure to Etoposide, alone or in combination with C2-4 or SSZ, no changes in ROS production were observed in HTLA-CSCs in respect of untreated CSCs. Instead, two-week treatment of ER-CSCs with Etoposide, alone or combined with C2-4 or SSZ, induced a 60% reduction of ROS levels in respect of untreated CSCs (Figure 9b). After five weeks of all treatments, no changes in ROS production were detected in ER-CSCs (Figure 9c).

Altogether these results demonstrate that although Etoposide alone significantly decreased the level of GSH, GSSG amount was not enhanced and, in parallel, ROS and MDA production did not increase. In this context, we cannot exclude that other antioxidant systems might contribute to the maintenance of CSC oxidative status [23,62] and in support of these findings, it has been recently found that the ability of breast CSCs to maintain low ROS production confers cancer stemness and drug resistance [63]. Interestingly, the co-treatments which are able to activate glycolysis, reduce GSH levels without altering cell redox state and induce lipid peroxidation are also effective in counteracting ER-CSC propagation.

### 3.6. Co-Treatments Are Able to Induce Down-Regulation of GPX4 Activity and of ZEB-1 Expression

Considering that xCT inhibition by SSZ or erastin has been demonstrated to induce lipid peroxide production and ferroptosis, a form of non-apoptotic cell death consequent to a loss of GSH and of Glutathione peroxidase 4 (GPX4) activity [64,65], the role of this enzyme was investigated.

As shown in Figure 10, after five weeks, in untreated ER-CSCs, the activity of GPX4 was reduced by 20% in respect of that observed after two weeks. Two-week-exposure to Etoposide, alone or in combination with SSZ, reduced the activity of GPX4 by 30% compared to that analyzed in untreated CSCs. Instead, two-week co-treatments with C2-4 further decreased GPX4 activity by 38% in respect of Etoposide alone and by 55% in comparison to the control. Moreover, after five weeks, Etoposide alone or in combination with C2-4 decreased GPX4 activity by 40% in respect of untreated CSCs, while SSZ co-treatments further inhibited the activity by 60% in comparison to Etoposide-treated CSCs and by 75% in respect of the control (Figure 10).

Since GPX4 plays a crucial role in the suppression of lipid peroxidation by stimulating a GSH-dependent reduction of phospholipid hydroperoxides [64,66], GPX4 inhibitors could be employed as anticancer drugs capable of inducing ferroptosis [67]. Interestingly, it has been reported that chemoresistant cancer cells, that are preserved from ferroptosis through GPX4 activation, expressed high levels of ZEB-1 [68]. ZEB-1 is a protein involved in the epithelial to mesenchymal transition (EMT) process, which plays a fundamental role in supporting CSC generation [69]. In fact, ZEB-1 has been found to have a pleiotropic role in controlling lipid metabolism, growth, proliferation and death of CSCs [68,70].

Based on these considerations and since Etoposide, alone or in combination with C2-4 or SSZ, was able to totally prevent ER-CSC formation, the subsequent analyses were focused on the role played by EMT in drug resistance.

As shown in Figure 11, both untreated HTLA- and ER-CSCs expressed N-Cadherin and ZEB-1, two proteins favoring the EMT process. Two-week-treatments of HTLA-CSCs fully counteracted the expression of these two proteins while, interestingly, two-week-Etoposide treatment of ER-CSCs reduced the expression of N-Cadherin and ZEB-1 by 40% and 70%, respectively, in comparison to untreated CSCs (Figure 11a,b). After five weeks, the combination of Etoposide with C2-4 reduced the expression of both proteins by 50% and co-treatment with SSZ decreased ZEB-1 levels by 90% in respect of untreated CSCs (Figure 11a,b).

Changes in the expression of N-Cadherin and ZEB-1 were not accompanied by significant alterations in the expression of Vimentin and β-catenin, another two EMT-related proteins (Figure 11c). In addition, as shown in Figure 11d, only treated HTLA-CSCs expressed Claudin, a protein inhibiting the EMT process.

Therefore, it is conceivable that co-treatments of Etoposide with SSZ/C2-4 are able to counteract the ability of ER-CSCs to generate spheres and to maintain the EMT phenotype. This data, although in vitro, supports the results obtained in early clinical trials demonstrating that the disruption of cancer stemness and/or the EMT process is a useful approach in fighting tumor recurrence [71,72].

Furthermore, in ER-CSCs, which become sensitive to the drug, GPX4 activity is reduced and is accompanied by ZEB-1 down-regulation leading to ferroptotic death [73,74].

The results obtained in co-treated CSCs confirm the ferroptotic-inducer action of SSZ [75] and suggest that C2-4, the PKCα inhibitor, is able to trigger ferroptosis of ER-CSCs. The relationship between PKCα and ferroptosis has been previously suggested in dopaminergic cells [76] but, to our knowledge, this is the first time it has been demonstrated that the anti-PKCα peptide can sensitize CSCs to chemotherapy by inhibiting GPX4 and inducing ferroptosis. PKCα activation has been implicated in promoting cancer progression [18,77] and in the formation and survival of cancer stem cells [78] Moreover, it has been demonstrated that the pharmacologic inhibition of PKCα can target breast CSCs [78] and override ZEB1-induced chemoresistance in hepatocarcinoma [79].

### 3.7. Chemoresistance of ER-CSCs Is Characterized by An Efficient OXPHOS Metabolism, High GSH Levels, ZEB-1 Up-Regulation and GPX4 Activation

In order to identify which variables are more determinant in maintaining cancer cell survival and, consequently, responsible for chemoresistance, Principal Component Analysis (PCA) was carried out by collecting all the analyzed metabolic, biochemical and morphological variables. The first two components explain 45.2% and 19.5% of the variance (64.6% in total).

The loading plot (Figure 12a) shows the correlation between the following variables: P/O ratio, ATP production, GSH and ZEB-1 expression and GPX4 activity. They are all characterized by negative loadings on PC1 (grouped in the green rectangle) and are all related since they allow the propagation of ER-CSCs. On the other side of the plot (positive loadings on PC1) another group of correlated variables can be detected (red rectangle): lactate release, glucose consumption and MDA production. They reduce ER-CSC propagation, and are responsible for their susceptibility to the co-treatments. The two groups of correlated variables are negatively correlated, since they are opposite compared to the origin (red cross in the plot). A third group of correlated variables can be detected, made by N-cad, OCR and Vim. These variables, having high loadings on PC2, are uncorrelated with the other ones.

As expected, the score plot (Figure 12b) shows that the two-week and five-week controls are close to each other, having negative PC1 scores. Therefore, they are characterized by the high values of the variables grouped into the green rectangle and the low values of the variables grouped into the red rectangle in the loading plot.

The effect of the three different treatments and of the different times, together with the metabolic modifications involved, can be estimated by taking into account the distance and the direction of the different points in the score plot, compared to the controls.

First of all, it can be seen that for all of them the effect after five weeks is larger than the effect after two weeks.

It can also be seen that the treatment with Etoposide by itself is much less efficient than the other two treatments in which Etoposide is combined with another drug. This treatment has almost no effect after two weeks, while after five weeks it leads to a more positive score on PC1, this meaning a reduction of the “green” variables and an increase of the “red” ones.

The co-treatment with SSZ has a much greater effect, already relevant after two weeks (corresponding to the effect after five weeks of Etoposide by itself). Since the direction is the same as the one already observed with Etoposide, it can be concluded that SSZ just amplifies the effect of Etoposide.

Instead, when looking at the co-treatment with C2-4 it can be seen that a strong effect is also present already at two weeks, but the trajectory followed is quite different. In this case an increase along PC1 together with a decrease along PC2 is observed, corresponding to a decrease of variables N-Cad, OCR and Vim. It can therefore be concluded that C2-4 not only amplifies the effect of Etoposide, but also produces different metabolic variations.

Collectively, these results further underline that CSC chemoresistance can be the result of a complex cellular adaptation that involves several metabolic and biochemical pathways and, therefore, it is necessary “to attack the enemy” simultaneously from several fronts in order to bypass this adaptation.

## 4. Conclusions

Even if the mechanism underlying CSC chemoresistance has not yet been fully clarified, the results herein reported suggest that drug refractoriness is maintained as long as CSCs are able to keep a fine balance between correlated molecular actors. In fact, the maintenance of an oxidative metabolism, high levels of GSH and the expression of ZEB-1 concur to allow the survival of resistant CSCs. Therefore, an approach able to down-regulate these molecular targets and to induce ferroptosis may be the winning strategy to counteract drug resistance. In this context, the results herein reported suggest that PKCα might be a new pharmacological target able to fight chemoresistance of cancer stem cells by preventing drug-induced metabolic adaptation and triggering ferroptotic death.

## Figures and Tables

**Figure 1 antioxidants-10-00691-f001:**
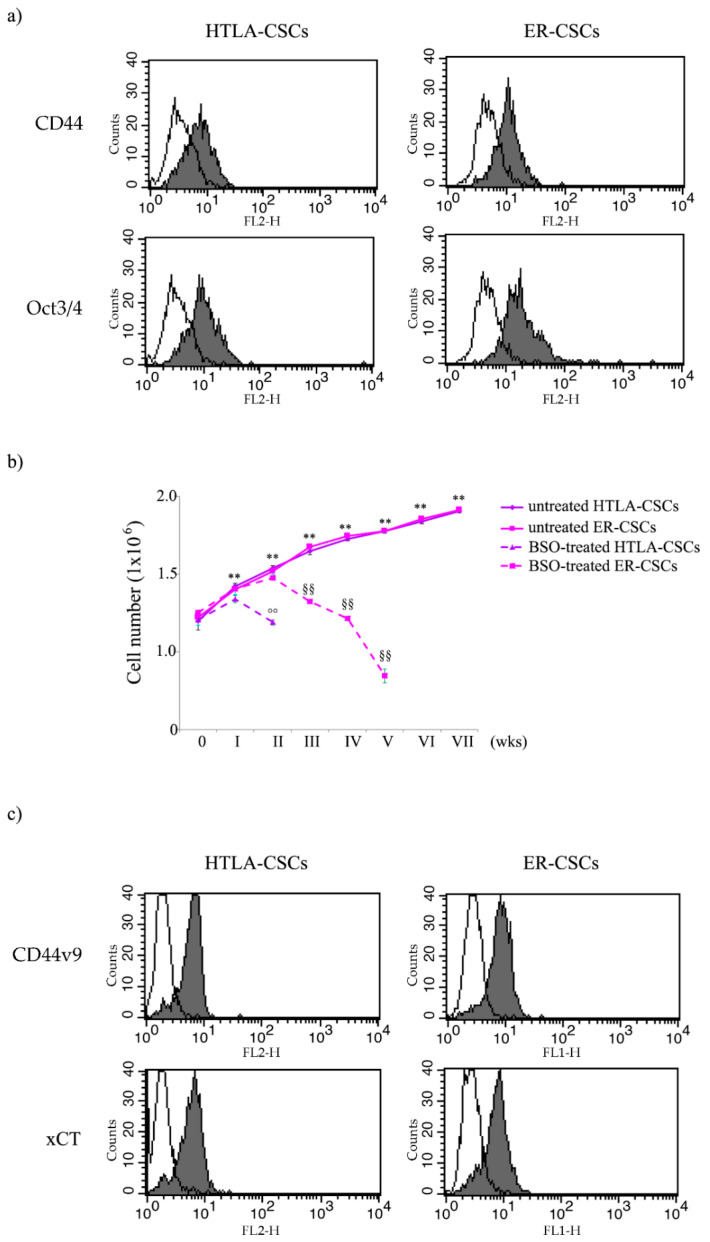
Flow cytometric analysis of CD44, Oct3/4 (**a**) and of CD44v9 and xCT (**c**) expression in untreated HTLA-CSCs and ER-CSCs. Cells were permeabilized, stained with mAbs to the indicated molecules, and analyzed by flow cytometry. Grey profiles indicate cells stained with the different mAbs, while white profiles correspond to isotype control. One representative experiment of five is shown. (**b**) Evaluation of cell number in HTLA-CSCs and ER-CSCs chronically treated with 1mM BSO. Both CSC populations were treated (once a week for six weeks) with 1 mM BSO and, at any split, disaggregated CSCs were counted using a Burker chamber as described in Materials and Methods. ** *p* < 0.01 vs. time 0; °° *p* < 0.01 vs. untreated HTLA-CSCs; ^§§^
*p* < 0.01 vs. untreated ER-CSCs.

**Figure 2 antioxidants-10-00691-f002:**
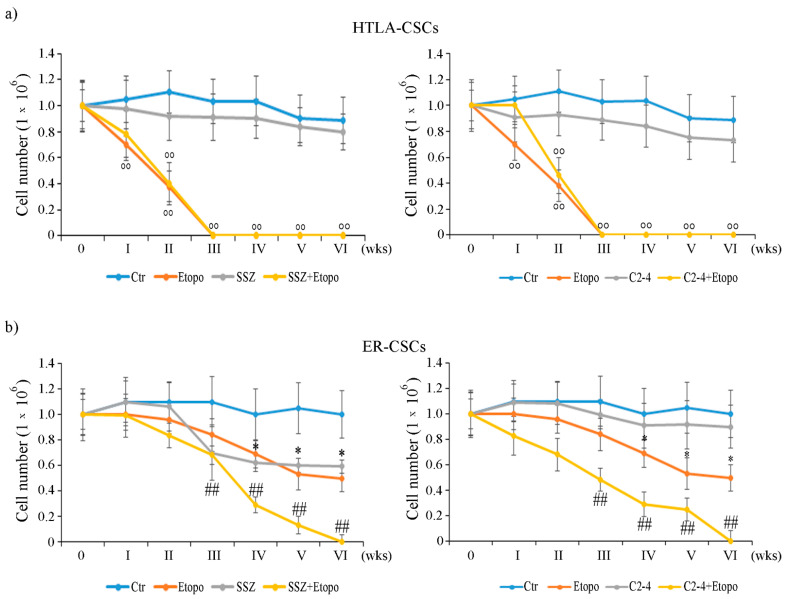
Evaluation of cell number in HTLA-CSCs (**a**) and ER-CSCs (**b**) treated three times a week with 0.1 µM C2-4, with 5 µM SSZ or with 1.25 µM Etoposide, administered once a week alone or in combination with C2-4 or SSZ. The treatments were protracted for six weeks. In order to evaluate CSC propagation, at any split, disaggregated CSCs were counted using a Burker chamber as described in Materials and Methods. °° *p* < 0.01 vs. untreated HTLA-CSCs (Ctr); * *p* < 0.05 vs. untreated ER-CSCs (Ctr); ^##^
*p* < 0.01 vs. Etoposide-treated ER-CSCs.

**Figure 3 antioxidants-10-00691-f003:**
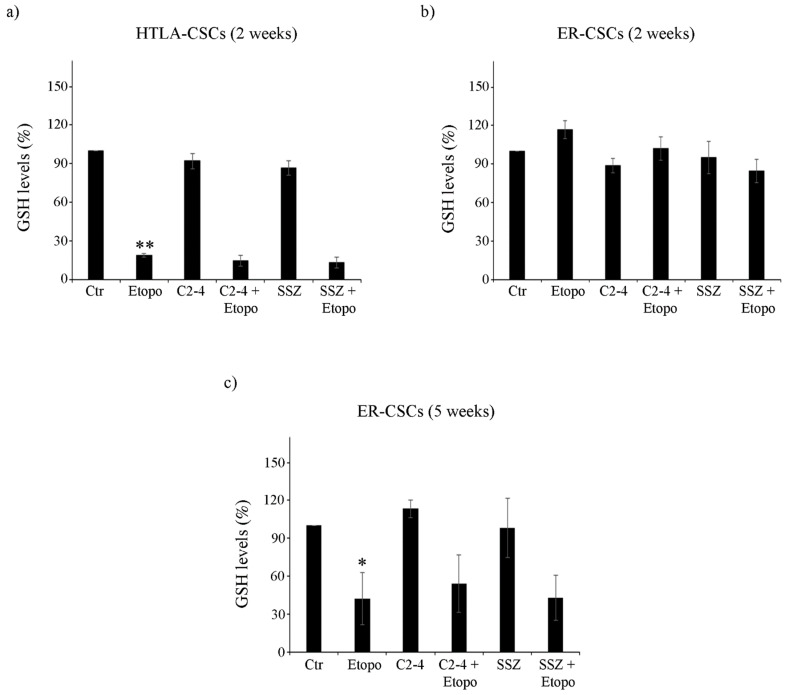
Glutathione (GSH) levels in both CSC populations treated for two (**a**,**b**) or five weeks (**c**). HTLA-CSCs and ER-CSCs were treated three times a week with 0.1 µM C2-4, with 5 µM SSZ or with 1.25 µM Etoposide, administered once a week alone or in combination with C2-4 or SSZ. GSH concentrations were determined by HPLC analysis and expressed as percentages of the control value. * *p* < 0.05 vs. untreated CSCs (Ctr); ** *p* < 0.01 vs. untreated CSCs (Ctr).

**Figure 4 antioxidants-10-00691-f004:**
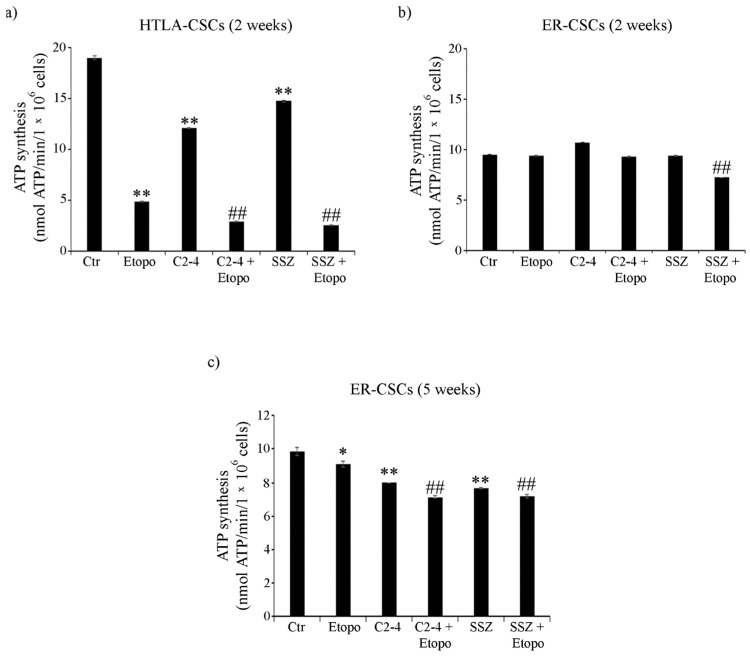
ATP synthesis in both CSC populations treated for two (**a**,**b**) or five weeks (**c**). HTLA-CSCs and ER-CSCs were treated three times a week with 0.1 µM C2-4, with 5 µM SSZ or with 1.25 µM Etoposide, administered once a week alone or in combination with C2-4 or SSZ. Results were reported as nmol ATP/min/10^6^ cells. Bar graph summarizes quantitative data of means ± S.E.M. of three independent experiments. * *p* < 0.05 vs. untreated CSCs (Ctr); ** *p* < 0.01 vs. untreated CSCs (Ctr); ^##^
*p* < 0.01 vs. Etoposide-treated CSCs.

**Figure 5 antioxidants-10-00691-f005:**
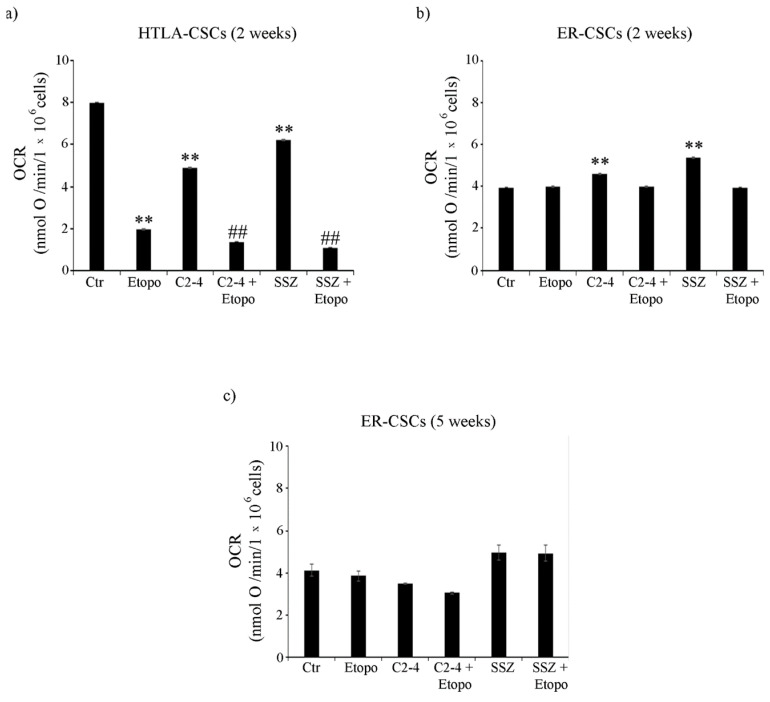
Oxygen consumption rate (OCR) in both CSC populations treated for two (**a**,**b**) or five weeks (**c**). HTLA-CSCs and ER-CSCs were treated three times a week with 0.1 µM C2-4, with 5 µM SSZ or with 1.25 µM Etoposide, administered once a week alone or in combination with C2-4 or SSZ. Results were reported as nmol O/min/10^6^ cells. Bar graph summarizes quantitative data of means ± S.E.M. of three independent experiments. ** *p* < 0.01 vs. untreated CSCs (Ctr); ^##^
*p* < 0.01 vs. Etoposide-treated CSCs.

**Figure 6 antioxidants-10-00691-f006:**
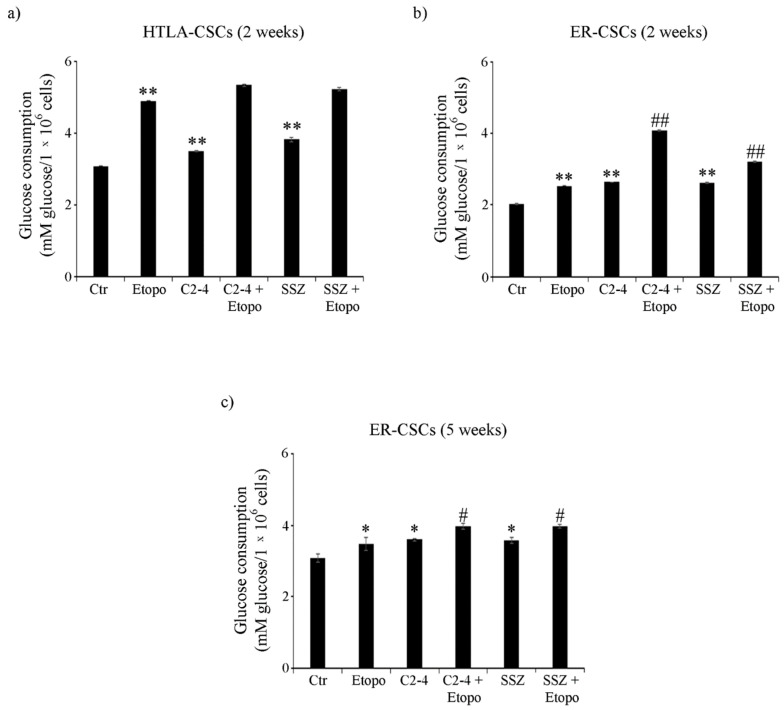
Glucose consumption in both CSC populations treated for two (**a**,**b**) or five weeks (**c**). HTLA-CSCs and ER-CSCs were treated three times a week with 0.1 µM C2-4, with 5 µM SSZ or with 1.25 µM Etoposide, administered once a week alone or in combination with C2-4 or SSZ. Results were reported as mM glucose/10^6^ cells. Bar graph summarizes quantitative data of means ± S.E.M. of three independent experiments. * *p* < 0.05 vs. untreated CSCs (Ctr); ** *p* < 0.01 vs. untreated CSCs (Ctr); ^#^
*p* < 0.05 vs. Etoposide-treated CSCs; ^##^
*p* < 0.01 vs. Etoposide-treated CSCs.

**Figure 7 antioxidants-10-00691-f007:**
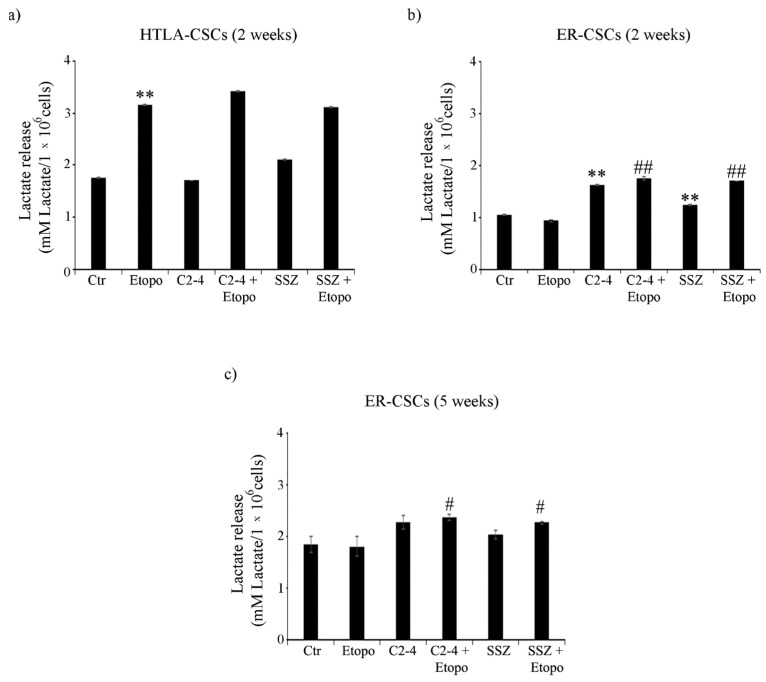
Lactate release in both CSC populations treated for two (**a**,**b**) or five weeks (**c**). HTLA-CSCs and ER-CSCs were treated three times a week with 0.1 µM C2-4, with 5 µM SSZ or with 1.25 µM Etoposide, administered once a week alone or in combination with C2-4 or SSZ. Results were reported as mM lactate/10^6^ cells. Bar graph summarizes quantitative data of means ± S.E.M. of three independent experiments. ** *p* < 0.01 vs. untreated CSCs (Ctr); ^#^
*p* < 0.05 vs. Etoposide-treated CSCs; ^##^
*p* < 0.01 vs. Etoposide-treated CSCs.

**Figure 8 antioxidants-10-00691-f008:**
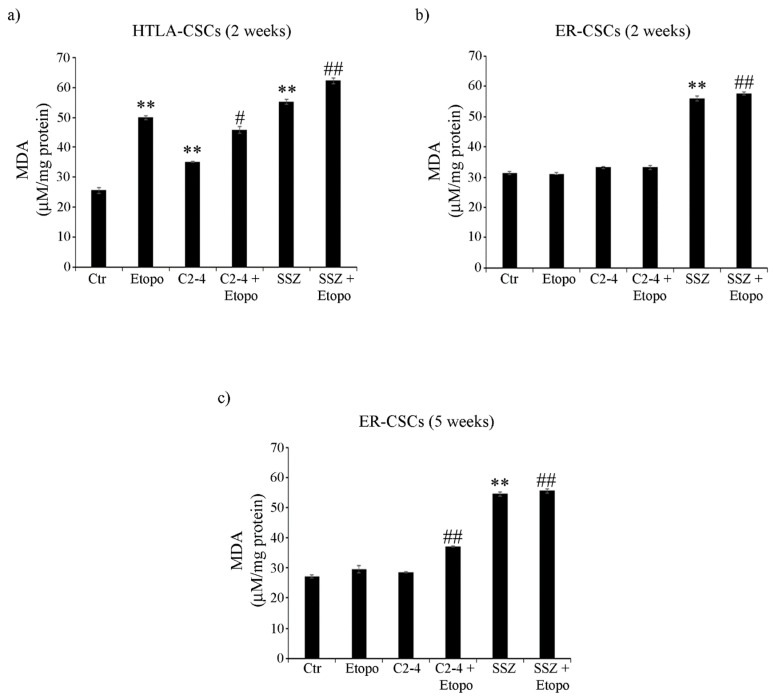
MDA production in both CSC populations treated for two (**a**,**b**) or five weeks (**c**). HTLA-CSCs and ER-CSCs were treated three times a week with 0.1 µM C2-4, with 5 µM SSZ or with 1.25 µM Etoposide, administered once a week alone or in combination with C2-4 or SSZ. Results were reported as μM/mg protein. Bar graph summarizes quantitative data of means ± S.E.M. of three independent experiments. ** *p* < 0.01 vs. untreated CSCs (Ctr); ^#^
*p* < 0.05 vs. Etoposide-treated CSCs; ^##^
*p* < 0.01 vs. Etoposide-treated CSCs.

**Figure 9 antioxidants-10-00691-f009:**
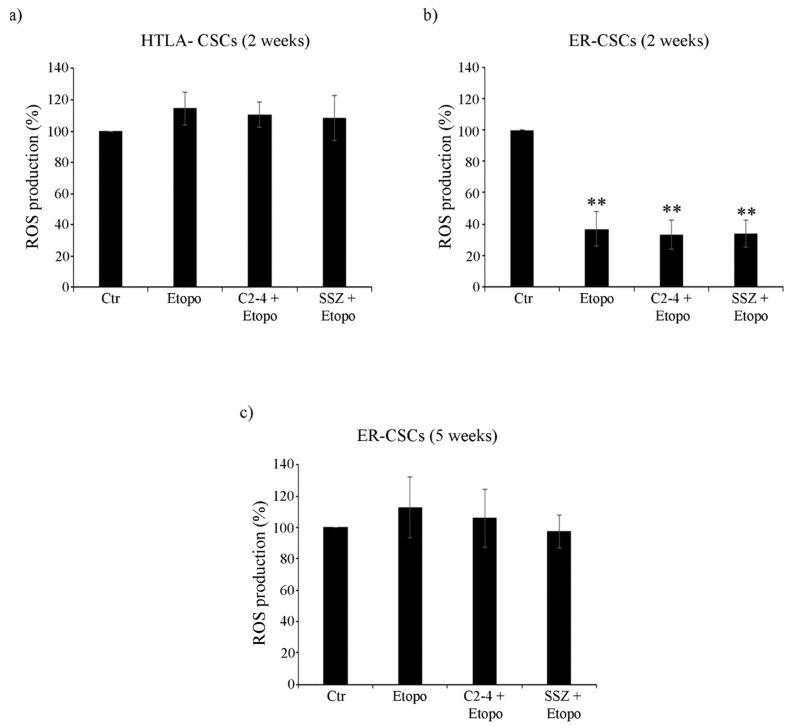
ROS production in both CSC populations treated for two (**a**,**b**) or five weeks (**c**). HTLA-CSCs and ER-CSCs were treated three times a week with 0.1 µM C2-4, with 5 µM SSZ or with 1.25 µM Etoposide, administered once a week alone or in combination with C2-4 or SSZ. Results were expressed as percentages of the control value. Bar graph summarizes quantitative data of means ± S.E.M. of three independent experiments. ** *p* < 0.01 vs. untreated CSCs (Ctr).

**Figure 10 antioxidants-10-00691-f010:**
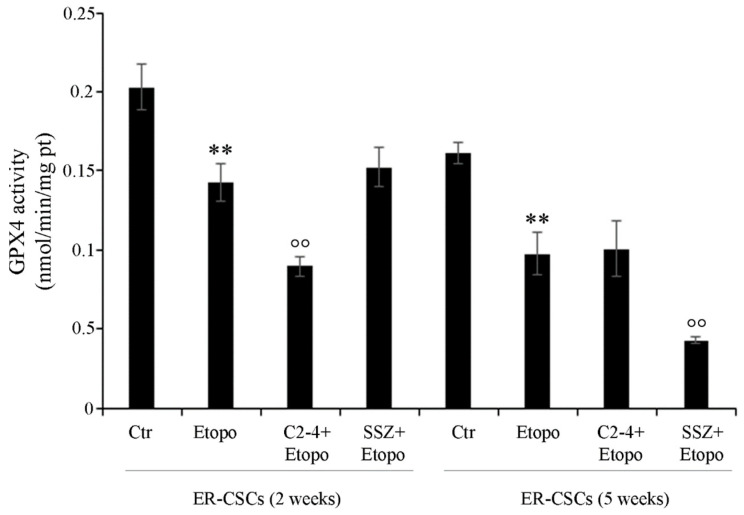
GPX4 activity in ER-CSC populations treated for two or five weeks. ER-CSCs were treated three times a week with 0.1 µM C2-4, with 5 µM SSZ or with 1.25 µM Etoposide, administered once a week alone or in combination with C2-4 or SSZ. Results were reported as nmol/min/mg protein. Bar graph summarizes quantitative data of means ± S.E.M. of three independent experiments. ** *p* < 0.01 vs. untreated ER-CSCs (Ctr); °° *p* < 0.01 vs. Etoposide-treated ER-CSCs.

**Figure 11 antioxidants-10-00691-f011:**
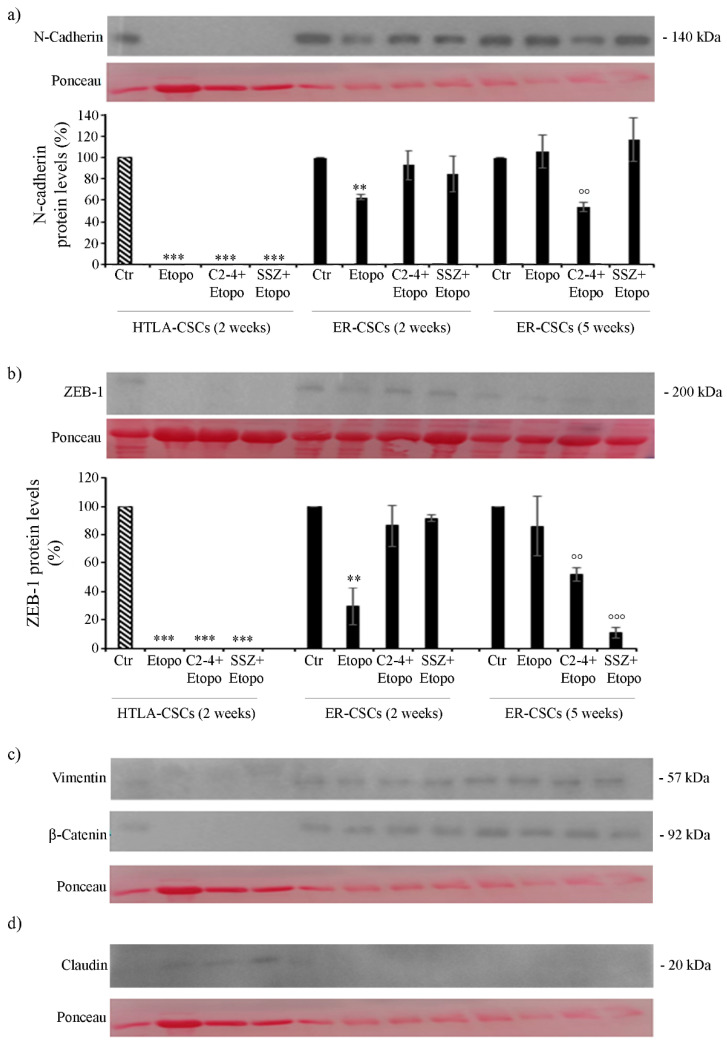
Protein levels of N-Cadherin (**a**), ZEB-1 (**b**), Vimentin (**c**), β-Catenin (**c**) and Claudin (**d**) in both CSC populations treated for two or five weeks. HTLA-CSCs and ER-CSCs were treated three times a week with 0.1 µM C2-4, with 5 µM SSZ or with 1.25 µM Etoposide, administered once a week alone or in combination with C2-4 or SSZ. Immunoblots shown are representative of three independent experiments. To ensure normalized protein content all filters were stained with Red Ponceau. Bar graph summarizes quantitative data of means ± S.E.M. of three independent experiments. ** *p* < 0.01 vs. untreated CSCs (Ctr); *** *p* < 0.001 vs. untreated CSCs (Ctr); °° *p* < 0.01 vs. Etoposide-treated CSCs; °°° *p* < 0.001 vs. Etoposide-treated CSCs.

**Figure 12 antioxidants-10-00691-f012:**
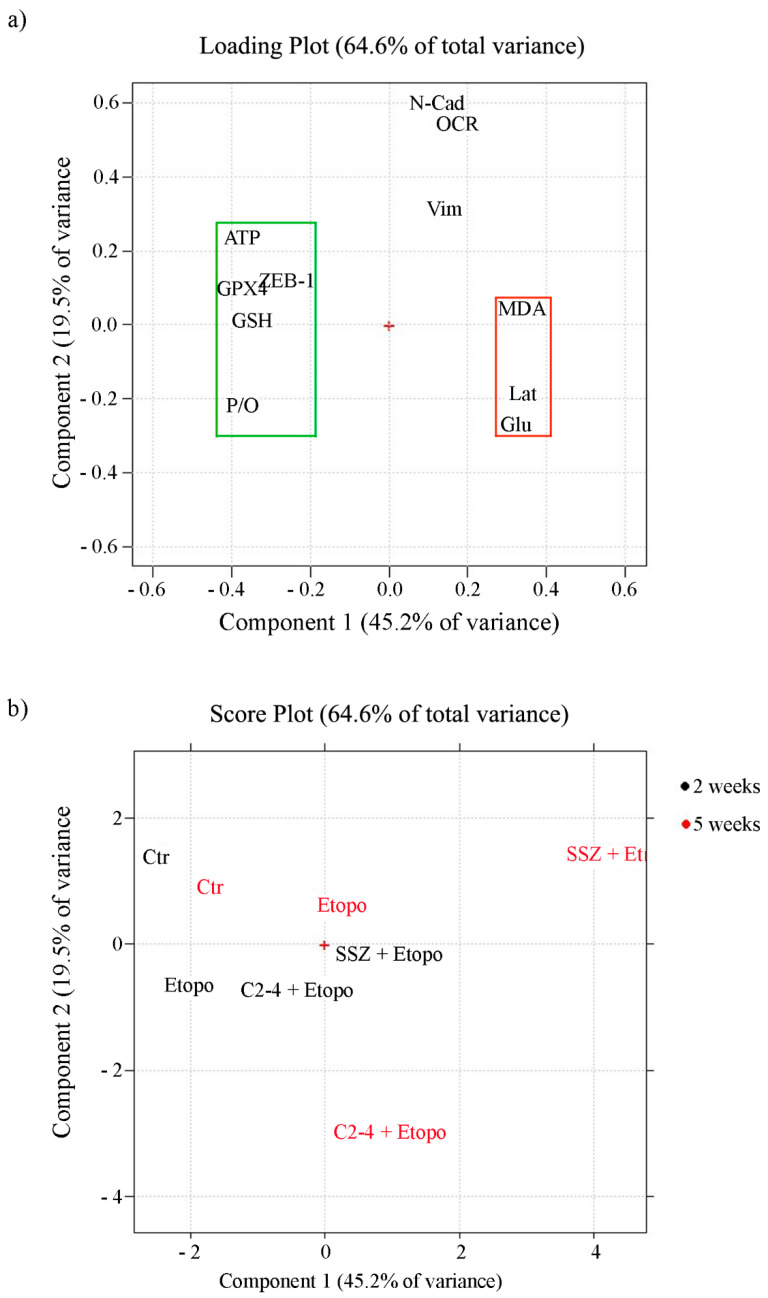
Loading (**a**) and score (**b**) plots. (**a**) The variables which sustain chemoresistance (green rectangle: ATP, GSH, P/O, ZEB-1, GPX4) have negative loadings on PC1 values, while the variables that are able to induce chemosensitivity (red rectangle: MDA production, lactate release and glucose consumption) have positive loadings on PC1; variables N-CAD, OCR and Vim have instead high loadings on PC2 (**b**) In the score plot, the black-colored samples were treated for two weeks, while the red-colored ones were treated for five weeks.

## Data Availability

Not applicable.

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
