# Peer review of "PKCα Inhibition as a Strategy to Sensitize Neuroblastoma Stem Cells to Etoposide by Stimulating Ferroptosis"

_antioxidants, 2021, doi:10.3390/antiox10050691_

Round 1
Reviewer 1 Report
An interesting study, the outcomes of which are very relevant to the field. Conclusions drawn are logical, and the paper has some very interesting and valid data. There are a number of points, some of which I'm sure are relatively simple fixes, others (Western data) might need more repeats to convince readers of the data.
1) The authors refer extensively to the cells as CSC, yet show no detail as to have they are different to the bulk population. The isolation of floating cells does yield cells with CSC markers, but what about the non-floating cells/bulk population?
2) Similarly, all of the effects noted are limited to the CSC, and the study does not inform us how the bulk population behaves. For example fig 1, what is the profile of the bulk population and what are the BSO sensitivities of the bulk? Are the observations unique to the CSC, and are the bulk population easily and effectively killed?
3) Methods does not have full required details. It would be helpful to have details on what concentration the drugs were diluted to in DMSO to give the reader some clues as to how to replicate (concentration in frozen/stored aliquots), and clearly articulate what does of DMSO is present in all experiments, ans these were referred to just as 'control'.
4) Methods: Some methods were absent with just a reference to a previous study [20]. A brief method with at least the suppliers of key components as a bare minimum. Some methods are modifications of previous methods that might not be in open access journals, although [20] at least is open access. ATP, glucose, lactate, MDA are those identified for more detail.
5) Antibody clone numbers required in western blotting section as some suppliers have more than one antibody for that particular protein.
6) Following on from point 1, neurospheres are first mentioned in the results. The cell lines and CSC population/characterisation/previous experience with these cells could well be in the introduction and or methods and would make the paper more coherent. Earlier mention of floating cells (line 89) does not immediately translate to neurospheres for most readers.
7) 3.2. "Etoposide prevents the formation of HTLA-CSCs... "I think this could be better worded, its really relating to proliferation. Formation suggests, well, formation of CSC (from something else), not maintaining cell numbers.
8) The Ponceau staining appears a little odd. Firstly, why show a very cropped ponceau section? Is that representative of all protein masses? Also for 11 a c and d the same loading has been used, whereas a different loading is for Zeb-1. Is there a scientific reason for this? The protein loading for the HTLA-CSCs is highly variable and shows evidence of unequal staining/transfer mid band in lane 5. The overloading correlates with absence of main targets on the western. Overall the Westerns are very unconvincing. The attempted alignment of the Westerns with the histograms was not intuitive at all. This is the weakest results section and strongly recommend amending/improving this section.
Author Response
An interesting study, the outcomes of which are very relevant to the field. Conclusions drawn are logical, and the paper has some very interesting and valid data. There are a number of points, some of which I'm sure are relatively simple fixes, others (Western data) might need more repeats to convince readers of the data.
We thank the Reviewer for the positive comments and appreciation.
1) The authors refer extensively to the cells as CSC, yet show no detail as to have they are different to the bulk population. The isolation of floating cells does yield cells with CSC markers, but what about the non-floating cells/bulk population?
We are sorry for the misunderstanding and we try now to clarify. The procedure used to in vitro select CSCs does not allow the survival of cancer cells without stem characteristics and, as shown in Fig. 1, the isolation of floating cells yield NB cells with CSC markers. Moreover, in our previous paper cited in the bibliography [28], the formation of 3D cultures or neurospheres from monolayer NB cells (2D cultures) is better described and, in this context, we demonstrated that neurospheres showed a major expression of stem cell markers in comparison with monolayer cells.
2) Similarly, all of the effects noted are limited to the CSC, and the study does not inform us how the bulk population behaves. For example fig 1, what is the profile of the bulk population and what are the BSO sensitivities of the bulk? Are the observations unique to the CSC, and are the bulk population easily and effectively killed?
Also this point needs to be better clarified. As reported by several studies, added in the bibliography of the revised manuscript [9-11], the presence of CSCs in the tumor bulk in vivo is responsible for the failure of chemotherapy. Therefore, the aim of this in vitro study is focused to investigate new approaches targeting CSC in order to fight NB chemoresistance.
However, in our previous paper [23] we reported that 24 hour-BSO treatment decreased the viability of monolayer HTLA-230 cells and sensitized them to Etoposide. Inversely, BSO did not modify the sensitivity of monolayer HTLA-ER cells to the cytotoxic drug.
3) Methods does not have full required details. It would be helpful to have details on what concentration the drugs were diluted to in DMSO to give the reader some clues as to how to replicate (concentration in frozen/stored aliquots), and clearly articulate what does of DMSO is present in all experiments, ans these were referred to just as 'control'.
As required, details about drugs’ dilution in DMSO have been added.
4) Methods: Some methods were absent with just a reference to a previous study [20]. A brief method with at least the suppliers of key components as a bare minimum. Some methods are modifications of previous methods that might not be in open access journals, although [20] at least is open access. ATP, glucose, lactate, MDA are those identified for more detail.
As suggested, the required details have been added in Materials and Methods section.
5) Antibody clone numbers required in western blotting section as some suppliers have more than one antibody for that particular protein.
In agreement with the Reviewer, the clone number for each antibody, used to perform western blotting analyses, has been added.
6) Following on from point 1, neurospheres are first mentioned in the results. The cell lines and CSC population/characterisation/previous experience with these cells could well be in the introduction and or methods and would make the paper more coherent. Earlier mention of floating cells (line 89) does not immediately translate to neurospheres for most readers.
We apologize for the lack of clarity on this issue. As mentioned in the point 1, the term neurospheres cited in the Results section is referred to the multicellular spheres obtained from the floating cells derived from 2D cultures. As described in Materials and Methods, only these floating cells are able to generate neurospheres (3D cultures) when grown in the DMEM-F12 Knock-out containing 1% penicillin/streptomycin, 2% B27, 40 ng/ml basal growth factor for fibroblasts (bFGF) and 20 ng/ml epidermal growth factor (EGF) [24]. It is important to outline that these neurospheres are called CSCs only after the characterization of stem cell markers. In addition, the term “floating cells” (line 89) is not referred to neurospheres but to single cells in suspension that originate from the monolayer of parental cells grown in the optimal conditions for the development of 2D cultures.
However, to avoid misunderstanding, this part has been better explained in Materials and Methods (see lines 89-98).
7) 3.2. "Etoposide prevents the formation of HTLA-CSCs... "I think this could be better worded, its really relating to proliferation. Formation suggests, well, formation of CSC (from something else), not maintaining cell numbers.
In agreement with the Reviewer’s suggestion, we explained in the Results section that treatment with Etoposide prevents the formation and propagation of CSCs. In fact, as described in Materials and Methods section, at any split CSCs were collected, centrifuged and dissociated to obtain a mono-disperse cell population from which CSCs originate. Treatments with drugs, alone or in combination, totally prevent the formation of HTLA-CSCs or ER-CSCs after 3 and 6 weeks respectively, and limit the propagation of the CSCs that are generated until 2 weeks (HTLA-CSCs) and 5 weeks (ER-CSCs).
8) The Ponceau staining appears a little odd. Firstly, why show a very cropped ponceau section? Is that representative of all protein masses? Also for 11 a c and d the same loading has been used, whereas a different loading is for Zeb-1. Is there a scientific reason for this? The protein loading for the HTLA-CSCs is highly variable and shows evidence of unequal staining/transfer mid band in lane 5. The overloading correlates with absence of main targets on the western. Overall the Westerns are very unconvincing. The attempted alignment of the Westerns with the histograms was not intuitive at all. This is the weakest results section and strongly recommend amending/improving this section.
Taking into consideration the Reviewer's comments we have tried to improve the quality of Figure 11 and to clarify the doubts.
The non cropped version of both Ponceau stainings, representative of all protein masses, have been reported in the “Supporting information file” submitted together with the paper following the Author’s guidelines. Therefore, we suppose that probably the Reviewers have not received this file during the revision.
As highlighted by the Reviewer, the same loading has been used for the panels a c and d of Fig. 11, whereas a different loading has been used for Zeb-1. The reason is why N-Cadherin, Vimentin, b-Catenin and Claudin have a molecular weight different enough to allow a good gel separation: in fact, by incubating the same membrane with each antibody, a precise detection of the specific protein was obtained. Instead, Zeb-1 has a molecular weight that could generate doubts when the same PVDF membrane is incubated with anti-N-Cadherin antibody and then with anti-Zeb-1, also after re-blotting. In fact, the degree of separation on gradient SDS-gel of proteins with high molecular weight is less marked in comparison with that is obtained for proteins with low molecular weight. Therefore, for this reason we carried out two different Western blots.
Reviewer 2 Report
In this study, the authors specifically investigate the role of PKCα in CSCs progression and its metabolism process during Etoposide treatment or other combination treatment. Good writing. However, there are some major weaknesses of this study that would have to be addressed before this acceptable for publication.
Major question
1.These CSCs derived from different parental cells, endogenous expression of stem cell marker, including CD44, Oct3/4 in these two cell lines (HTLA-CSCs and ER-CSCs) should be explored.
2.Follwoing the question 1, the expression of CD44v9 and xCT should be addressed in HTLA-CSCs and ER-CSCs, that might help us easier to connect the further experiments.
3. GSH metabolism is a major feature of ongoing oxidative stress in chemotherapeutic drugs treatment of cancer cells. To more specifically speculate GSH metabolism, the GSH/GSSG ratio should be addressed in figure 3.
In addition, the level of GSH (GSH/GSSG ratio) and ROS production are associated with each other. Comparing to figure 3c and 9c, Etoposide significantly decreased the level of GSH but not increased ROS production, why? Other antioxidant components involved in this process? This part should address in discussion.
4.According to the result from figure 4-7, indicating that ER-CSC especially under co-treatment C2-4 or SSZ with Etoposide shifted their metabolic process into glycolytic metabolism, thus glycolysis may serve as a protective role in ER-CSC upon Etoposide or combination treatment. 2-DG or other glycolysis inhibitor could be applied in ER-CSS model.
5.C2-4 or SSZ combined with Etoposide synergistically induced ferroptosis in ER-CSCs. To further demonstrate these results, ferroptosis inhibitor, such as ferrostatin-1 should be utilized in these models.
6. Figure 11 should be reorganized.
For example, internal control marker, Ponceau do not use the same band, or these bolts should be reorganized as “one figure”.
The bands in figure 11 was not sharp, do you have better bands?
There is only mesenchymal marker in figure 11, how about epithelial marker? E-cadherin.
Author Response
In this study, the authors specifically investigate the role of PKCα in CSCs progression and its metabolism process during Etoposide treatment or other combination treatment. Good writing. However, there are some major weaknesses of this study that would have to be addressed before this acceptable for publication.
We thank the Reviewer for the positive comments and appreciation.
Major question
- These CSCs derived from different parental cells, endogenous expression of stem cell marker, including CD44, Oct3/4 in these two cell lines (HTLA-CSCs and ER-CSCs) should be explored.
We are sorry but we don’t understand this request since the endogenous expression of both CD44 and Oct3/4 has been evaluated in permeabilized HTLA-CSCs and ER-CSCs, as reported in Materials and Methods, and the results of flow cytometric analysis were shown in Figure 1a.
- Following the question 1, the expression of CD44v9 and xCT should be addressed in HTLA-CSCs and ER-CSCs, that might help us easier to connect the further experiments.
Analogously to question 1, the endogenous expression of both CD44v9 and xCT was evaluated and reported in Figure 1c.
- GSH metabolism is a major feature of ongoing oxidative stress in chemotherapeutic drugs treatment of cancer cells. To more specifically speculate GSH metabolism, the GSH/GSSG ratio should be addressed in figure 3.
We absolutely agree with the Reviewer that GSH/GSSG ratio better describes the role of GSH metabolism in the acquisition of chemoresistance. However, in our model, the evaluation GSH/GSSG ratio was not possible since, as described in the Results section, GSSG levels were below to the detection limits in both CSC populations after 2 weeks of treatments.
In addition, the level of GSH (GSH/GSSG ratio) and ROS production are associated with each other. Comparing to figure 3c and 9c, Etoposide significantly decreased the level of GSH but not increased ROS production, why? Other antioxidant components involved in this process? This part should address in discussion.
As already reported in the original manuscript, our results demonstrate that the survival of Etoposide-resistant CSCs is not totally dependent on GSH since, even though Etoposide exerts a severe GSH-depleting action, it does not totally counteract CSC generation, suggesting that other factors might contribute to the maintenance of cancer stem cell survival. Moreover, as suggested by the Reviewer, although Etoposide significantly decreased the level of GSH, GSSG amount was not enhanced and in parallel ROS production did not increase. Given that the co-treatments able to counteract CSC survival decreased GPX4 activity and enhanced lipoperoxidation, we have better discussed this issue in the text.
- According to the result from figure 4-7, indicating that ER-CSC especially under co-treatment C2-4 or SSZ with Etoposide shifted their metabolic process into glycolytic metabolism, thus glycolysis may serve as a protective role in ER-CSC upon Etoposide or combination treatment. 2-DG or other glycolysis inhibitor could be applied in ER-CSS model.
The results reported in Figure 4-7, indicate that as a long as CSCs are able to maintain an efficient OXPHOS metabolism, they are able to grown and only when treatments induce a shift to glycolysis, CSCs generation and propagation is counteracted. For this reason, we believe that the use of 2-DG or of other glycolysis inhibitors cannot help us to sustain the results of our research. On the contrary, in agreement with our study it has been demonstrated that OXPHOS inhibitors (e.g. metformin) have been tested in combination with chemotherapeutic drugs in order to increase therapy sensitivity of cancer stem cells [60,61].
- C2-4 or SSZ combined with Etoposide synergistically induced ferroptosis in ER-CSCs. To further demonstrate these results, ferroptosis inhibitor, such as ferrostatin-1 should be utilized in these models.
We agree with the Reviewer, but considering that the selection of CSCs would take about 2 months of experimental work, we cannot perform the experiments suggested in ten days. Notably, given SSZ has been widely recognized as an inducer of ferroptosis [74,75] we can indirectly support the ferroptotic action of C2-4. In fact, all markers of ferroptosis such as GSH depletion, lipid peroxidation, reduction of GPX4 activity and ZEB-1 down-regulation, were found in both co-treatments with SSZ- or C2-4. However, these findings deserve to be further investigated and the experiments suggested by the Reviewer will be the objective of the next future study.
- Figure 11 should be reorganized.
For example, internal control marker, Ponceau do not use the same band, or these bolts should be reorganized as “one figure”.
The bands in figure 11 was not sharp, do you have better bands?
Taking into consideration the Reviewer's comments we have tried to improve the quality of Figure 11 and to clarify the doubts.
The non cropped version of both Ponceau stainings, representative of all protein masses, have been reported in the “Supporting information file” submitted together with the paper following the Author’s guidelines. Therefore, we suppose that probably the Reviewers have not received this file during the revision.
There is only mesenchymal marker in figure 11, how about epithelial marker? E-cadherin.
We agree with the Reviewer that is necessary to show both mesenchymal and epithelial markers and accordingly we have evaluated and reported the expression of Claudin, a known epithelial marker.
Round 2
Reviewer 2 Report
The authors well response all questions. The acceptation is suggested.